# The Evaluation of Active Learning Classrooms: Impact of Spatial Factors on Students' Learning Experience and Learning Engagement

**Lei Peng** [1,2]**, Yuan Deng** [1,2] **and Shitao Jin** [1,2,*]

1 School of Architecture & Urban Planning, Huazhong University of Science and Technology, Wuhan 430074, China; penglei@hust.edu.cn (L.P.); m202073637@hust.edu.cn (Y.D.)
2 Hubei Engineering and Technology Research Center of Urbanization, Wuhan 430074, China
* Correspondence: m202073616@hust.edu.cn; Tel.: +86-189-8601-1150

**Abstract:** Previous studies have investigated the spatial attributes of Active Learning Classrooms (ALCs) and their impact on students' learning experiences and learning engagement independently; however, a holistic investigation of the relationship between these attributes and students' learning engagement has not been conducted. This study explored how the spatial attributes of ALCs affect students' learning experiences and learning engagement. An empirical questionnaire was administered to freshmen taking English classes in four different types of ALCs at one university, and 224 valid questionnaires were analyzed. This study provides design insight for future learning environments in ALCs by linking two Likert scales: one rating spatial attributes in ALCs that influence learning experiences, and the other rating students' learning engagement in ALCs. The results revealed that the spatial attributes of ALCs significantly affected the learning experience, specifically instructional interaction, furniture perception, learning support, and physical environment. Among them, instructional interactions and physical environment are the most critical in promoting student learning engagement. The survey findings can help architects design more flexible and sustainable learning environments in the future, supporting university students in developing active and collaborative learning skills, as well as providing better references and beneficial insights for future education for sustainable development.

**Keywords:** active learning classrooms; learning environments; spatial attribute; learning engagement; learning experience; sustainable development

## 1. Introduction

This paper presents the results of a study on the spatial factors that influence college students' learning experiences and their relationship to learning engagement in Active Learning Classrooms (ALCs). The emergence of ALCs in universities has bettered the original classroom space, advocating the use of an innovative teaching and learning model of the information age to improve students' learning effectiveness, guiding them to switch from passive learning to active learning, and from superficial learning to deep learning. The researchers found that students in ALCs outperformed aptitude-based expectations in terms of learning outcomes when compared to students in traditional classrooms [1,2]. As a means of putting Education for Sustainable Development (ESD) processes into practice, ALCs have received great attention for their ability to create intelligent, personalized, and adaptive learning environments [3]. Currently, the practice and research of ALCs in various countries have emerged as a critical subject in global education development.

The concept and practice of ESD have been major driving forces in the establishment and development of ALCs during the last two decades. UNESCO led and coordinated the Decade of Education for Sustainable Development (abbr. UNDESD, 2005–2014) in

2005, emphasizing the integration of ESD principles into all levels of education, meaning that education for sustainable development necessitates a holistic shift in classroom space, teaching techniques, and learning styles [4]. In 2015, UNESCO re-emphasized in its Education 2030 Framework for Action that ESD necessitates a rethinking of the physical learning space in light of sustainable development, with a learning environment integrated with a variety of digital devices and learning software to support better learning and development for teachers and students [5]. It is confirmed that ALCs act as a transformative educational methodology allowing students to acquire problem solving and critical thinking skills, as UNESCO stated in its learning objectives in education for sustainable development goals (SDGs) [6].

ALCs, as a product of ESD, are designed to optimize active learning for students [7]. This classroom model combines architectural design and active learning pedagogy with a variety of flexible furniture and technological equipment to enhance student learning [8]. Radcliffe's "Pedagogy-Space-Technology" framework [9] proposed in "Next Generation Learning Space" has been widely used in the theoretical study of ALCs. The significance of this framework is that it considers the relationship between pedagogy, space, and technology in ALCs as an organic whole instead of isolated units. A large portion of theoretical research on ALCs is devoted to pedagogy; for example, Sawers et al. ascertained that the more that teachers used active teaching strategies in ALCs, the better they could use ALCs to increase student engagement [10]. Basdogan et al. found that student engagement increased when teachers made good use of their instructional strategies with space and technology in ALCs [11]. Spatial studies on ALCs have focused on classroom layout. In a quasi-experimental study, Byers et al. observed that students studying in ALCs had significantly higher motivation and attitudes toward learning than students studying in traditional classroom layouts [12]. Odum et al. found that multiple spatial layouts of ALCs had a positive impact on student learning engagement [13]. Several studies and practices have been conducted on the technology of ALCs. For example, Xiaohai et al. applied image recognition technology to ALCs, which can real-time track the classroom content, serve as technical equipment to assist teaching, and provide an efficient recording service for the subsequent development of teaching activities [14]. Hasan et al. proposed a framework for using video streaming servers and forecasting techniques to enhance the teaching and learning process in ALCs [15].

Exploring ALCs' practices began in the 1990s with North Carolina State University's Student-Centered Active Learning Environment for Undergraduate Program (SCALE-UP), which reversed the teacher-centered approach to a student-centered environment [16]. SCALE-UP also uses technological devices to improve classroom operability and flexibility, allowing students to interact and collaborate in small groups while directly accessing the content [17]. MIT's Technology Enables Active Learning (TEAL), developed in 2003, is a prototype that helps students get a visual comprehension of curriculum concepts and principles to enhance attendance and lower student failure rates. TEAL's Innovation can be seen in more advanced visual media simulations and individual response systems to improve student collaboration and learning [18]. The PAIR-up (Pedagogy-rich; Assess learning impact; Integrate innovations; Revisit emerging technologies) model, based on SCALE-UP and TEAL, was proposed by the University of Minnesota in 2006. This model utilized the most popular wall system technology at the time, using demountable walls and spliceable floor materials, significantly increasing ALCs' flexibility [19]. TILE was proposed by the University of Iowa in 2012, and it is distinguished by combining classroom space and a variety of teachers' teaching strategies. The classroom meets the specialized needs of each discipline, so TILE is also known as "SCALE-UP" that can be transferred among multiple disciplines [20].

Even though diverse spatial layouts and ever-richer technological equipment have led to significant changes in ALCs, the following questions remain to be answered. Which spatial factors can influence students' learning experiences? Which spatial factors can promote students' learning engagement? This study distributed an empirical questionnaire

to students enrolled in ALCs at Huazhong University of Science & Technology (HUST) to collect students' learning experience and learning engagement indicators, and aimed to answer the following research questions:

1. What spatial elements influence students' learning experiences in ALCs?
2. Are there differences in the effects on students' learning experiences and learning engagement in different types of ALCs? What spatial elements contribute to the variation in these differences?
3. What major spatial elements of ALCs increase student participation in learning? How may they be improved further?

## 2. Theoretical Background

One of the critical mediating variables for measuring students' learning effectiveness in ALCs is their learning engagement. This theoretical background is derived from ALCs' spatial layout and students' learning engagement, summarizes current domestic and international research findings, and forms the analytical framework for subsequent empirical studies accordingly.

### 2.1. Summary of Spatial Elements of ALCs

Several researchers have investigated the relationship between the spatial elements of ALCs and students' learning experiences to evaluate the use of ALCs. They have proposed some design principles and a set of critical features of ALCs. This study summarized four spatial elements of ALCs that primarily impact learning experiences, and developed a subsequent learning experience survey scale based on these four spatial elements using established research literature.

Numerous studies have found that the physical environment in learning spaces influences the learning experience and learning effectiveness. Many empirical studies have found that natural and artificial light sources in ALCs are essential factors influencing students' learning experiences [21]. Good natural light sources meet students' basic physiological and psychological needs, and softer artificial lighting environments significantly improve students' learning comfort [22]. Some researchers have already measured the relationship between the degree of temperature variation and student learning in the ALCs during different seasons. The results show that a comfortable temperature and humidity environment can improve students' learning experience [23]. The ALC's interior environment is also a critical factor influencing the student learning experience; for example, at Michigan Technological University, the walls and furniture were chosen in earth tones, with wooden table surfaces, warm green chairs, and sunset orange hues on the walls. This natural tone interior environment accords a fully enhanced student learning experience [22].

Compared to traditional classrooms, the spatial layout of ALCs has a significant impact on students' learning experiences. Through quasi-experiments and questionnaires, Byers et al. ascertained that the spatial layout of ALCs positively affect students' learning experiences and motivation more than the spatial layout of traditional classrooms [12]. Smith argued that the geometry of different classroom spatial layouts affects their internal hierarchy, which affects teachers' and students' psychological ownership of the space, and that the spatial layout of the service center truly moves toward students only when the geometry of the space inhibits symmetry and potentially axisymmetric shaping [24]. The usable area per capita is also the design focus of the space layout; Vincent J et al. concluded in their article that the classroom layout with a large usable area per capita is the most satisfactory for students; the classroom space with a fixed arrangement of tables and chairs leads to a lower utilization rate of students [21]. However, classroom space that is too spacious also has some disadvantages, such as the difficulty of students in hearing other students [25].

In addition to the spatial layout, ALCs' furniture design can also directly impact the learning experience. Numerous studies have found that features such as 360-degree swivel seats [26], seats with rollers [27], U-shaped collaborative desks [28], and furniture with

integrated tables and chairs [29] significantly impact student learning in ALCs. However, some researchers have mentioned a few flaws about the furniture in ALCs; for example, Robert et al. evaluated improvements to the design of the flat armchair in the ALCs at the University of North Carolina at Chapel Hill, which students barely used, owing to the small size of the slanted shelf under the seat and its tendency to fall off when moving [27].

Information technology devices have significantly altered the way teaching and learning occur in ALCs, and greatly impact the students' learning experience. Numerous studies have shown that electronic display devices in ALCs positively impact student classroom interactions [30]. Diogo et al. proposed that ALCs provide more projector screens in different configurations or use projector screen display devices with curved shapes to improve student learning [31]. A few studies have emphasized the importance of interactive whiteboards [32,33]. Though the technology content of whiteboards varies between ALCs that have them, mobile whiteboards provide students with plenty of space for communication and interaction, which has a positive and productive impact on their learning experience [34].

### 2.2. The Influence of ALCs' Spatial Factors on Student Learning Engagement

In an ALC learning environment, students can participate more actively in the classroom than in traditional classrooms; however, there are also distractions, laziness, and other low participation behaviors. Students in China are still accustomed to listening and learning in traditional lecture-based classrooms rather than classroom collaboration and discussion, owing to the relatively short period of Chinese ALCs development. It is necessary to conduct systematic research on the influencing factors of student engagement and their mechanisms of action in ALCs, to improve students' engagement and, thus, active learning ability. This paper summarizes and synthesizes the literature on the factors influencing student engagement in ALCs, focusing on behavioral, affective, and cognitive dimensions.

ALCs are distinguished by more diverse modes of teaching and learning, which bring more frequent interactive behaviors between teachers and students, or students and students. For example, more space for teachers and students to move around the classroom [26], and more diverse modes of collaboration among students [35], have increased student behavioral engagement to varying degrees. McArthur concluded that students' learning behaviors are more diverse in ALCs after comparing classroom participation in traditional classrooms and ALCs [36]. Furthermore, some researchers have reported that the ALC learning environment significantly impacts how teachers teach and students learn, and positively impacts student behavioral engagement [37]. However, some researchers have noted that overly convenient furniture and equipment can distract students, and cause interferences between individual student behaviors [38].

Student emotional engagement in ALCs has improved, owing to the student-centered design philosophy. ALCs for cooperative learning also enrich students' peer emotions [36]. Information technology devices allow students to quickly switch between independent and group learning [30]. Furthermore, the ALCs' learning environment improved students' sense of belonging, and influenced their affective engagement [39]. Some researchers concluded that students' emotional engagement levels do not change significantly in ALCs compared to traditional classrooms, owing to the different classroom locations [40]. According to Smith, the irregularity of the spatial layout of classrooms in ALCs contributes to the high level of student emotional engagement [24]. According to the study by Young et al., eye contact in ALCs is essential for student emotional engagement, and future ALCs designs should emphasize the importance of visual communication [41].

The primary goal of ALCs is to promote active and deep learning, which necessitates a higher level of cognitive engagement to comprehend complex concepts, master more difficult skills, and develop self-management skills. ALCs can provide an interdisciplinary learning environment that encourages a multidisciplinary and interprofessional approach to learning while also improving students' multi-metacognitive skills [20]. Sawers et al. discovered higher levels of student engagement in ALCs, and a degree of influence of the

instructor's instructional strategies on student cognitive engagement [10]. McDavid et al. proposed that the teacher's teaching style influenced students' cognitive engagement in traditional classrooms [42]. In ALCs, there was no significant correlation between changes in students' cognitive engagement and changes in teaching strategies. Other researchers cited the effective use of space in ALCs and technological equipment as essential factors in student cognitive engagement [43].

### 2.3. Conceptual Model

Based on the research presented above, this study conceptualizes and operates the spatial elements of ALCs that influence students' learning experiences into four dimensions: physical environment, spatial layout, furniture design, and technological equipment; and divides students' learning engagement in ALCs into three dimensions: behavioral dimension, emotional dimension, and cognitive dimension. This study investigates the critical spatial factors that influence learning experience, compares the effects of different ALCs on learning experience and learning engagement, and investigates the relationship between spatial factors and learning engagement in ALCs, as shown in the figure (Figure 1).

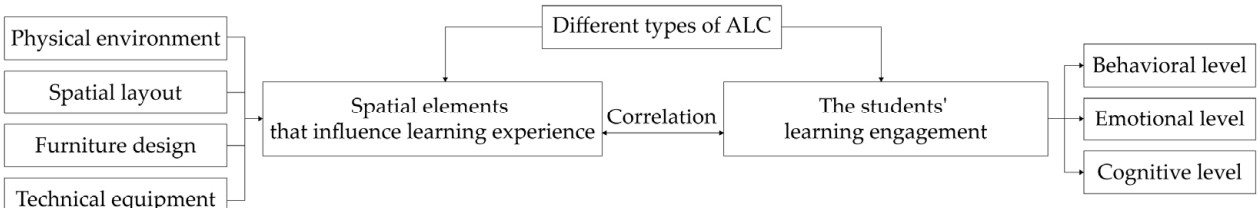

**Figure 1.** Research conceptual framework.

## 3. Survey Methodology

### 3.1. Survey Design

The study was conducted at the ALCs at Huazhong University of Science and Technology (HUST), and a questionnaire research method was employed (see Appendix A). The following are the reasons for the selection: First, this university's ALCs have been in use since 2018, a period of four years, and the number of ALCs exceeds 110, which is representative in terms of sample size. Second, the ALCs at this university carry out spatial layout according to SCALE-UP design principles, and make specific technical updates and furniture design according to the development of the times and educational needs, resulting in four distinct modes (Figure 2): ALCs (individual) of splice combination type, ALCs of table and chair integrated type, ALCs (multiple people) of splice combination type, and ALCs (multiple people) of fixed combination type. ALCs from various categories enrich the spatial elements, and serve as comparative references for subsequent differentiation analysis.

The subjects of this study were first-year college students who had spent one semester at the ALCs. The course chosen was a comprehensive required course: General English, which included various students from various majors. This course is entirely taught in ALCs, and all classes are taught in small sizes. The study period was from 2020 to the second half of the 2021 semester, with 56 credit hours (4 credit hours per week) for one semester. They encouraged the students to use the different spatial layouts and technological devices in the ALCs. The course is taught to ensure that students experience the different spatial layouts and technological devices in the classroom.

Before the start of that English course, the instructor asked students to complete a general English knowledge proficiency test, dividing each class into 4 to 5 groups of 5 to 7 students, each based on test scores and gender ratio, to eliminate factors of variation in learning ability and gender differences in each class group. A questionnaire completion process was carried out at the end of this semester, and the questionnaires were distributed over two weeks.

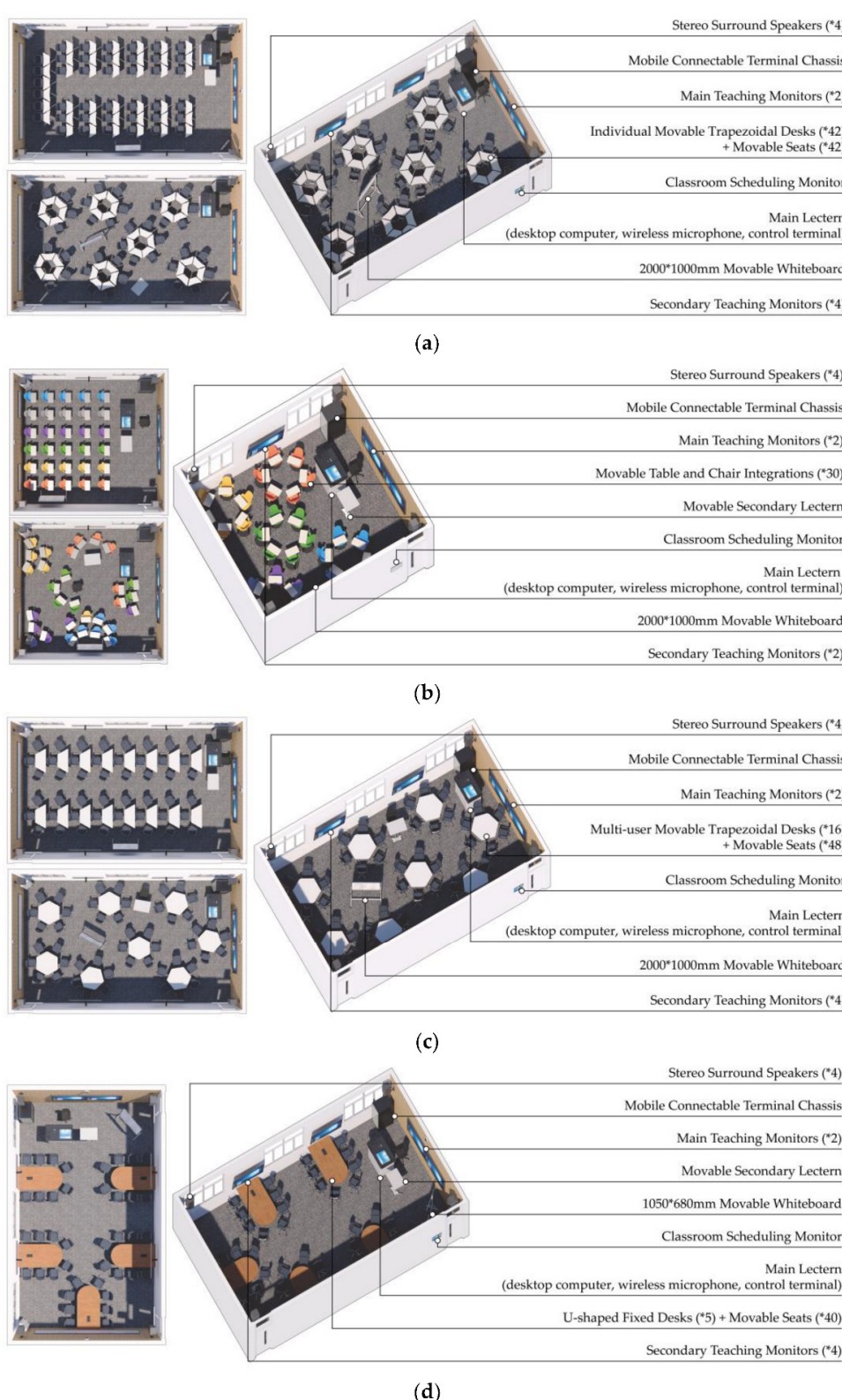

**Figure 2.** Four types of ALCs in HUST: (**a**) ALCs (individual) of splice combination type; (**b**) ALCs of table and chair integrated type; (**c**) ALCs (multiple people) of splice combination type; (**d**) ALCs (multiple people) of fixed combination type.

### 3.2. Questionnaire Design

The "Spatial Influences on Students' Learning Experience in ALCs" scale was created using the four dimensions summarized in the literature review, including the physical en-

vironment dimension, which was inspired by the "Investigation of Environmental Quality (IEQ) in Classrooms" scale designed by Choi et al. [44]. The questions for the spatial layout dimension were based on the "Influence Scale on Student Performance in Active Learning Classroom Environments" developed by Yang et al. [45]. The furniture design dimension was based on the "Student Classroom Seating Rating Scale (CSRS-S)" designed by Harvey et al. [46]. The technological equipment dimension was based on the Active Learning Classroom Environment Assessment Survey Scale developed by the University of Minnesota's Active Learning Assessment Team [19]. Each dimension's questions were also written separately in conjunction with the different types of ALCs in HUST. The final scale included 18 items, including six questions about the physical environment, four questions about spatial layout, two questions about furniture design, and six questions about technological equipment, and the response options were provided on a 5-point Likert scale.

The "Students' engagement in learning in ALCs" scale was based on the three dimensions summarized in the literature review concerning Wang's mature scale of "Chinese college students' engagement in learning questionnaire" [47]. The final scale set up 17 questions, including seven questions of behavioral dimension, four questions of emotional dimension, and six questions of cognitive dimension, and the response options were provided a 5-point Likert scale.

## 4. Results

### 4.1. Status and Exploratory Factor Analysis

The statistical software SPSS 24 was used as the primary tool for analyzing the questionnaires. In this study, the two scales were combined into a single questionnaire for distribution, and 246 questionnaires were collected. After excluding incomplete, duplicate, or invalid responses, 224 valid questionnaires were used for analysis. The scale "Spatial Influences on Students' Learning Experience in ALCs" had a reliability of 0.95, and the scale "Students' Learning Engagement in ALCs" had a reliability of 0.97.

Students perceived the spatial aspects of the ALCs to be quite influential in their learning experience, as shown in Table 1, with a mean value of 4.25. Among these variables, students ranked the "spatial layout" and "furniture design" as the most influential dimensions, followed by the "physical environment" dimension, and the "technological equipment" as the least influential dimensions. Students' learning engagement in ALCs was relatively high, as shown in Table 2, with a mean value of 4.148. The emotional dimension had the highest level of student engagement, followed by the behavioral dimension, and the cognitive dimension had the lowest level of student engagement.

**Table 1.** Statistics on the current state of spatial factors influencing students' learning experiences in ALCs.

|  | Number | Min | Max | Mean | SD | Variance |
|---|---|---|---|---|---|---|
| Physical environment | 224 | 1.33 | 5.00 | 4.2426 | 0.72944 | 0.532 |
| Space layout | 224 | 1.00 | 5.00 | 4.2813 | 0.78604 | 0.618 |
| Furniture design | 224 | 1.00 | 5.00 | 4.2813 | 0.90534 | 0.820 |
| Technical equipment | 224 | 1.00 | 5.00 | 4.2292 | 0.80393 | 0.646 |
| Overall | 224 | 1.11 | 5.00 | 4.2510 | 0.72346 | 0.523 |

All data were obtained by descriptive statistics of the questionnaire conducted by SPSS.

**Table 2.** Statistics on the current state of students' learning engagement in ALCs.

|  | Number | Min | Max | Mean | SD | Variance |
|---|---|---|---|---|---|---|
| Behavioral dimension | 224 | 1.00 | 5.00 | 4.1480 | 0.77763 | 0.605 |
| Emotional dimension | 224 | 1.00 | 5.00 | 4.1622 | 0.76984 | 0.593 |
| Cognitive dimension | 224 | 1.25 | 5.00 | 4.1272 | 0.78237 | 0.612 |
| Overall | 224 | 1.06 | 5.00 | 4.1481 | 0.72872 | 0.531 |

All data were obtained by descriptive statistics of the questionnaire conducted by SPSS.

An exploratory factor analysis of the scale "Spatial Influences on Students' Learning Experience in ALCs" produced a KMO value of 0.917, which is greater than 0.9, indicating an excellent level, and Bartlett's spherical test with $p = 0.000 < 0.05$. After several explorations, three items were removed to yield four factors with eigenvalues greater than 1, accounting for 74.064% of the explainable variance. According to the initial dimensional design of the scale, factor 1 includes the content related to teaching resources, interactive devices, and teacher-student activities in ALCs, so factor 1 can be named as "instructional interaction." Factor 2 includes furniture design and environmental perception in ALCs, so factor 2 can be named "furniture perception." Factor 3 includes learning services and humanized facilities in ALCs, so factor 3 can be named "learning support." Factor 4 includes the acoustic-, optical-, and thermal-environment-related contents in ALCs, so factor 4 can be named as "physical environment." Each factor was replaced by "F1", "F2", "F3", and "F4", and the re-clustered and named factors are shown in Table 3. Since the "Students' Learning Engagement in ALCs" scale is a direct change from the validated scale, the dimensions of the scale remain the same.

**Table 3.** Spatial factors of ALCs after re-clustering and naming.

| Dimension | | Title Item | 1 | 2 | 3 | 4 |
|---|---|---|---|---|---|---|
| F1 | Instructional interaction | Multiple monitors are installed in the classrooms to allow for the presentation of course content from various angles. | 0.689 | | | |
| | | Multiple chalkboards (or whiteboards) make learning and communication more effortless in the classroom. | 0.632 | | | |
| | | The classroom layout promotes a more equal relationship between teachers and students. | 0.592 | | | |
| | | Interactive software is used in the classroom to help with teaching and learning. | 0.580 | | | |
| | | The classroom layout can be changed at any time to accommodate different teaching activities. | 0.557 | | | |
| F2 | Furniture perception | The classroom furniture is comfortable. | | 0.777 | | |
| | | Individual study areas are sufficiently large. | | 0.757 | | |
| | | The temperature in the classroom is suitable (warm in winter and cool in summer). | | 0.730 | | |
| | | The classroom furniture is flexible. | | 0.664 | | |
| F3 | Learning support | Classrooms can use their computers, etc., to facilitate electronic learning. | | | 0.792 | |
| | | The classroom has a good WIFI signal. | | | 0.688 | |
| | | There are plenty of power outlets in the classroom. | | | 0.670 | |
| F4 | Physical environment | The classrooms' natural light is sufficient. | | | | 0.752 |
| | | The classrooms' ventilation is adequate. | | | | 0.721 |
| | | The classrooms' sound insulation is adequate. | | | | 0.596 |

Extraction method: principal component analysis. Rotation method: Kaiser normalized maximum variance method. The rotation has converged after 16 iterations.

### 4.2. Differential Analysis of the Effects of 4 Types of ALCs

A single-factor ANOVA was used to see if different types of ALCs had different effects on students' learning experiences and engagement in learning. All four ALCs patterns were more than 0.05 in the variance test of both scales, demonstrating that the variance met the statistical requirements. Table 4 shows that there were no significant differential effects of different ALCs on any of the dimensions of students' learning experience; Table 5 shows that there were no significant differential effects of different ALCs on any of the dimensions of students' learning engagement.

**Table 4.** Comparison of the differences of each spatial dimension of the four types of ALCs on the learning experience.

| Test Variables | | Sum of Squares | Degrees of Freedom | Mean Sum of Squares | F-Test | Significance |
|---|---|---|---|---|---|---|
| | Intergroup | 2.789 | 3 | 0.930 | 1.671 | 0.174 |
| F1 | Intra-group | 122.426 | 220 | 0.556 | | |
| | Total | 125.215 | 223 | | | |
| | Intergroup | 2.523 | 3 | 0.841 | 1.118 | 0.343 |
| F2 | Intra-group | 165.509 | 220 | 0.752 | | |
| | Total | 168.031 | 223 | | | |
| | Intergroup | 3.251 | 3 | 1.084 | 1.482 | 0.220 |
| F3 | Intra-group | 160.844 | 220 | 0.731 | | |
| | Total | 164.095 | 223 | | | |
| | Intergroup | 1.450 | 3 | 0.483 | 0.728 | 0.536 |
| F4 | Intra-group | 146.034 | 220 | 0.664 | | |
| | Total | 147.484 | 223 | | | |
| | Intergroup | 2.316 | 3 | 0.772 | 1.485 | 0.220 |
| Overall | Intra-group | 114.400 | 220 | 0.520 | | |
| | Total | 116.716 | 223 | | | |

F1 is "instructional interaction"; F2 is "furniture perception"; F3 is "learning support"; and F4 is "physical environment".

**Table 5.** Comparison of the differences between the four types of ALCs for each dimension of learning engagement.

| Test Variables | | Sum of Squares | Degrees of Freedom | Mean sum of Squares | F-Test | Significance |
|---|---|---|---|---|---|---|
| | Intergroup | 0.886 | 3 | 0.295 | 0.485 | 0.693 |
| R1 | Intra-group | 133.965 | 220 | 0.609 | | |
| | Total | 134.851 | 223 | | | |
| | Intergroup | 1.414 | 3 | 0.471 | 0.793 | 0.499 |
| R2 | Intra-group | 130.748 | 220 | 0.594 | | |
| | Total | 132.162 | 223 | | | |
| | Intergroup | 0.532 | 3 | 0.177 | 0.287 | 0.835 |
| R3 | Intra-group | 135.967 | 220 | 0.618 | | |
| | Total | 136.499 | 223 | | | |
| | Intergroup | 0.893 | 3 | 0.298 | 0.557 | 0.644 |
| Overall | Intra-group | 117.529 | 220 | 0.534 | | |
| | Total | 118.422 | 223 | | | |

R1 is "behavioral dimension"; R2 is "emotional dimension"; R3 is "cognitive dimension".

*4.3. Regression Analysis of Spatial Factors and Learning Engagement*

A correlation analysis between the four spatial factors in the ALCs that affect the learning experience and the three dimensions of students' learning engagement revealed significant correlations, as shown in Table 6.

Multiple linear regression analysis was used to determine the importance of different spatial factors on students' learning engagement—four spatial factors affecting learning experience as independent variables and learning engagement as dependent variables. Because independent sample t-tests on students' gender and discipline revealed that neither gender nor major was a variable influencing students' engagement, no control variables were established for this linear regression.

The linear regression model fit well, as shown in Table 7, with $R^2 = 0.438$, indicating that the four factors affecting learning experience explained 43.8 percent of the variance in student engagement in learning. The regression equation was: "learning engagement" = 1.306 + 0.481 × "instructional interaction" + 0.291 × "physical environment". The independent variables are not multicollinear, and all VIFs are less than 5. Among the four spatial dimensions of ALCs, the factors "instructional interaction" and "physical environment" can have a significant impact on students' learning engagement (Beta = 0.494, $p < 0.001$; Beta = 0.325, $p < 0.001$).

**Table 6.** Comparison of correlations between the spatial elements of ALCs and the dimensions of learning engagement.

|    | F1 | F2 | F3 | F4 | IV | R1 | R2 | R3 | DV |
|----|----|----|----|----|----|----|----|----|----|
| F1 | 1 | | | | | | | | |
| F2 | 0.798 ** | 1 | | | | | | | |
| F3 | 0.764 ** | 0.643 ** | 1 | | | | | | |
| F4 | 0.703 ** | 0.664 ** | 0.636 ** | 1 | | | | | |
| IV | 0.953 ** | 0.889 ** | 0.845 ** | 0.819 ** | 1 | | | | |
| R1 | 0.574 ** | 0.406 ** | 0.470 ** | 0.557 ** | 0.567 ** | 1 | | | |
| R2 | 0.615 ** | 0.494 ** | 0.520 ** | 0.596 ** | 0.623 ** | 0.846 ** | 1 | | |
| R3 | 0.538 ** | 0.388 ** | 0.472 ** | 0.465 ** | 0.524 ** | 0.770 ** | 0.827 ** | 1 | |
| DV | 0.617 ** | 0.461 ** | 0.520 ** | 0.585 ** | 0.614 ** | 0.949 ** | 0.954 ** | 0.899 ** | 1 |

**. At the 0.01 level (two-tailed), the correlation is extremely significant; F1 is "instructional interaction"; F2 is "furniture perception"; F3 is "learning support"; F4 is "physical environment"; IV (independent variable) is "overall learning experience"; R1 is "behavioral dimension"; R2 is "emotional dimension"; R3 is "cognitive dimension"; and DV (dependent variable) is "overall learning engagement".

**Table 7.** Linear regression between each spatial factor of ALCs and learning engagement.

| Variable | B | SE | Beta | T | Sig |
|----------|---|----|------|---|-----|
| (Constant) | 1.306 | 0.224 | | 5.822 *** | 0.000 |
| F1 | 0.481 | 0.100 | 0.494 | 4.799 *** | 0.000 |
| F4 | 0.291 | 0.067 | 0.325 | 4.333 *** | 0.000 |
| R = 0.662 | $R^2 = 0.438$ | Adjusted $R^2 = 0.428$, F = 42.722 *** | | | |

***: $p < 0.001$; Variables: (constant); F1 for "instructional interaction"; F4 for "physical environment"; Dependent variable: learning engagement.

Furthermore, multiple linear regression analysis of the four spatial factors of ALCs and the three dimensions of students' learning engagement was used to determine the relationship between the spatial factors of ALCs and the students' behavioral, affective, and cognitive dimensions. Table 8 shows the results, and the $R^2$ for the three linear regressions were 0.396, 0.435, and 0.320, suggesting that the three linear regression models fit well. According to the findings, the "instructional interaction" component significantly impacts students' behavioral, emotional, and cognitive dimensions, whereas the "physical environment" factor significantly impacts students' behavioral and emotional dimensions.

**Table 8.** Linear regression between each spatial factor of ALCs and the three dimensions of learning engagement.

| Variable | B | SE | Beta | T | Sig |
|----------|---|----|------|---|-----|
| (Constant) | 1.312 | 0.248 | | 5.286 *** | 0.000 |
| F1 | 0.529 | 0.111 | 0.510 | 4.774 *** | 0.000 |
| F4 | 0.336 | 0.074 | 0.351 | 4.517 *** | 0.000 |
| R = 0.629 | $R^2 = 0.396$ | Adjusted $R^2 = 0.385$, F = 35.922 *** | | | |
| Dependent variable: Behavioral Dimension (R1). | | | | | |
| (Constant) | 1.151 | 0.238 | | 4.838 *** | 0.000 |
| F1 | 0.424 | 0.106 | 0.413 | 3.998 *** | 0.000 |
| F4 | 0.315 | 0.071 | 0.333 | 4.429 *** | 0.000 |
| R = 0.659 | $R^2 = 0.435$ | Adjusted $R^2 = 0.424$, F = 42.084 *** | | | |
| Dependent variable: Emotional Dimension (R2). | | | | | |
| (Constant) | 1.530 | 0.265 | | 5.771 *** | 0.000 |
| F1 | 0.480 | 0.118 | 0.460 | 4.059 *** | 0.000 |
| R = 0.565 | $R^2 = 0.320$ | Adjusted $R^2 = 0.307$, F = 25.736 *** | | | |
| Dependent variable: Cognitive Dimension (R3). | | | | | |

***: $p < 0.001$; Variables: (constant); F1 for "instructional interaction"; F4 for "physical environment".

## 5. Discussion

This study used the Huazhong University of Science and Technology's ALCs as the research object. The data collected were subject to statistical analysis to investigate, quan-

titatively, the spatial elements of the ALCs that affect students' learning experience and engagement. The findings revealed that "instructional interaction", "furniture perception", "learning support", and "physical environment" in ALCs were the most important factors that influenced students' learning experience. There was no statistically significant difference between the various types of ALCs in terms of students' learning experience and learning engagement. Finally, "instructional interaction" and "physical environment" factors in ALCs were found to effectively promote students' learning engagement.

### 5.1. In ALCs, "Teaching Interaction" and "Physical Environment" Are the Critical Spatial Factors That Promote Learning Engagement

The rapid advancement of information technology has resulted in the development of new approaches and ideas for the interaction mode and teaching methods of ALCs. The interactive discussion using whiteboards and multi-tool display media provides new educational teaching and learning support. The interaction characteristics change the original teaching methods and learning activities to create a better spatial environment for learning participation. Many studies have found that interactive whiteboards and multi-angle displays in ALCs effectively direct students' attention and engagement, bringing them closer to the content [22,31,33,48]. This study's findings validate the role of interactive devices in ALCs, and reinforce the role of this factor in facilitating student engagement.

Furthermore, intelligent interactive software that can be seamlessly integrated accelerates and enriches classroom content. Software that serves teacher–student interaction or student–student interaction could eliminate student distraction and inattention. Several studies have also found that student engagement increases when teachers' instructional strategies make good use of their software devices [11,38,49,50]. Those studies also suggest that interactive software devices can influence teaching styles and student learning, and make a significant difference in student engagement; however, training teachers on how to use the devices is also essential.

The equal spatial relationship between teachers and students and the flexible and versatile classroom layout are also essential factors in ALCs' ability to promote learning engagement. Teachers can walk among students without the confines of a podium, bringing them closer. Teachers are no longer just knowledge transmitters, and students are now active participants in their learning [26]. Flexible and adaptable spatial layouts influence students' attitudes toward their learning experiences and participation [12], and diverse spatial patterns, such as independent, group, and overall layouts, can better meet teachers' teaching strategies, and thus promote higher pedagogical gains in classroom learning.

This study, like others [21,25,44,51], confirms that improving the physical attributes of the learning environment improves student engagement. ALCs have more usable space per person and more student-centered classroom space than traditional classrooms, adjustable central air conditioning and artificial light increase control over environmental attributes, and provide users with various needs. Furthermore, acoustic insulation wool or wooden grills decorate the walls for increased sound insulation, and the focused and quiet environment encourages participation in learning.

### 5.2. Different Types of ALCs Have No Significant Impact on Students' Learning Experience and Learning Engagement

According to this study, there were no significant differences in the effects of the four types of ALCs on students' learning experience and learning engagement. However, the differences in the effects of "furniture perception" and "instructional interaction" dimensions on students' learning experience were, relatively, the greatest among the four ALC types. This may be owing to the different types of ALCs, outfitted with different furniture and interactive gadgets, resulting in varying spatial layouts, which impact the learning experience of students who utilize the furniture and devices. This study's findings are consistent with those of Walczak et al., who noted: "while students' learning experiences and learning engagement differed between ALCs, they all agreed that ALCs with varying furniture arrangements were successful in aiding the learning experience." [52].

Students' emotional engagement varied the most, owing to changes in classroom type. Though the values were not statistically significant, it was evident that changes in classroom space elements were the most likely to alter students' emotional engagement. Odum et al. discovered in their study that, although it was not possible to determine whether there were differences in student engagement findings across spatial layout formats, there was a relative increase in student engagement among the ALCs' classroom spatial layouts, demonstrating that the spatial layout format of the ALCs had a positive impact on student engagement in learning [13].

*5.3. The Positive Role of Furniture Design Factors in ALCs Should Be Continuously Researched and Expanded*

Although the "furniture perception" factor had no significant effect on learning engagement in this study, many empirical studies have found that furniture in ALCs can have a positive effect on students [27,30,52]. Contrarily, some studies have concluded that furniture in ALCs has some negative effect [35,38,53], so a follow-up in-depth study on this is required. The following are some possible explanations for the current study's findings: First, owing to Chinese university students' limited exposure to ALCs, they are still accustomed to selecting traditional teacher furniture configurations that are more suitable for independent learning, whereas the primary design principle of ALCs furniture is to promote group cooperation and patchwork combination working. This design pattern is diametrically opposed to the furniture arrangement that students have grown accustomed to in primary school, middle school, and high school, resulting in a non-significant correlation between the furniture perception dimension and students' learning engagement in the ALCs. Second, there is a limit to the investment and budget cost associated with ALCs in China, as well as a lack of investment and attention to furniture design in ALCs, which should be prioritized in future development. In their study, Zimmermann et al. also stated that "some of the unique furniture and equipment in ALCs may induce distractions and inattention in students. This problem will be rectified as ALCs grow more popular and evolve." [29].

## 6. Conclusions and Limitations

With the evolution and improvements in teaching models, space design, and technology configuration, ALCs have had a tremendous impact on the learning environment of ESD after more than 20 years of theoretical and practical development. Comparatively, the concept of ESD continues to improve and refine the design principles of ALCs [3], making the learning environment in ALCs more autonomous, inclusive, collaborative, and sustainable.

In this study, based on the literature review, we divided the spatial elements of ALCs that affect students' learning experience into four dimensions: physical environment, spatial layout, furniture design, and technological equipment, and divided students' learning engagement in ALCs into three dimensions: behavioral dimension, emotional dimension, and cognitive dimension. By distributing questionnaires to students in ALCs on the campus of HUST, and collecting and analyzing students' Likert scales of learning experience and learning engagement in ALCs, we obtained that all four spatial attributes of instructional interaction, furniture perception, learning support, and physical environment in ALCs significantly affect the learning experience, with instructional interaction and physical environment being the critical spatial factors that promote students' learning engagement. Based on this result, we can provide more targeted and sustainable recommendations for the future development of ALCs:

- Placing more flexible, general-purpose interactive devices in classrooms, such as mobile whiteboards and writable mobile displays, to facilitate different types of collaborative tasks and learning activities in which students can write, present, and report their ideas and knowledge;

- Creating a more flexible and accessible classroom space, allowing students to collaborate in any location within the classroom, and using a distributed layout to bring teachers and students closer together to create a more equal learning environment;
- Ensuring that students' learning spaces have optimal environmental conditions, such as acoustic-insulated doors and windows, regulation devices that dynamically combine natural and artificial lighting, and air conditioning equipment that improves air quality and temperature conditions, all of which should be adjustable and controllable in real-time to make a more efficient use of energy.

These spatial elements are rethought and redesigned in order to provide more flexible and sustainable learning environments that enable students to make more effective use of teaching and learning resources, and an innovative use of learning equipment and furniture. These redesigned ALCs will improve students' learning experience and promote more effective learning engagement, allowing students to engage in real ESD processes, and help them to develop sustainable active learning skills.

Furthermore, this study has certain flaws, owing to our restricted competence in the actual study and the influence of research time. For example, the study solely used questionnaires to assess learning experiences and learner engagement. It lacked process data collection and analysis, such as classroom observations or qualitative analysis of teachers and students. Therefore, they can be incorporated in future studies (the original data is in the supplementary material).

**Supplementary Materials:** The following are available online at https://www.mdpi.com/article/10.3390/su14084839/s1.

**Author Contributions:** L.P. conceived and designed the research, and critically reviewed the article; S.J. wrote the paper; Y.D. and S.J. analyzed the data. All authors have read and agreed to the published version of the manuscript.

**Funding:** This research was funded by the National Natural Science Foundation of China, grant number 51978294.

**Institutional Review Board Statement:** The study was conducted according to the guidelines of the Declaration of Helsinki, and approved by the Medical Ethics Committee, Tongji Medical College, Huazhong University of Science and Technology.

**Informed Consent Statement:** Informed consent was obtained from all subjects involved in the study.

**Data Availability Statement:** The data used to support the findings of this study are included within the article.

**Acknowledgments:** The authors would like to thank the English teachers and freshmen at Huazhong University of Science and Technology for their support and encouragement.

**Conflicts of Interest:** The authors declare no conflict of interest.

## Appendix A. Questionnaire Template

*Appendix A.1. The Scale of Spatial Factors Influencing Students' Learning Experiences in ALCs*

Hello! Thank you very much for taking the time to fill out this questionnaire; this research aims to understand your learning experience in the ALCs. Your feedback will help to update and improve the quality of classroom space. Thank you!

1. Your gender is:

        A. Male                 B. Female

2. Your major is:

  A. Philosophy, Economics, and Law
  B. Education, Literature, and History
  C. Science, Engineering, Agriculture, and Medicine
  D. Military Science, Management, and Art

3. Your classroom number for the course "General English" this semester:
4. Do you agree that the following factors help to improve your learning experience when you attend classes in ALCs?

| [1 = Strongly Disagree; 2 = Disagree; 3 = Neutral Attitude; 4 = Agree; 5 = Strongly Agree] | | | | | |
|---|---|---|---|---|---|
| **Title** | **1** | **2** | **3** | **4** | **5** |
| Physical Environment | | | | | |
| The classroom's sound insulation is adequate. | ☐ | ☐ | ☐ | ☐ | ☐ |
| The classroom's natural light is sufficient. | ☐ | ☐ | ☐ | ☐ | ☐ |
| The classroom's artificial lighting is adequate. | ☐ | ☐ | ☐ | ☐ | ☐ |
| The temperature in the classroom is suitable (warm in winter and cool in summer). | ☐ | ☐ | ☐ | ☐ | ☐ |
| The classroom's ventilation is adequate. | ☐ | ☐ | ☐ | ☐ | ☐ |
| The classrooms are nicely decorated. | ☐ | ☐ | ☐ | ☐ | ☐ |
| Space Layout | | | | | |
| The classroom has a clear view that is not obstructed by other furniture or equipment. | ☐ | ☐ | ☐ | ☐ | ☐ |
| The classroom layout can be changed at any time to accommodate different teaching activities. | ☐ | ☐ | ☐ | ☐ | ☐ |
| The classroom layout promotes an equal relationship between teachers and students. | ☐ | ☐ | ☐ | ☐ | ☐ |
| Individual study areas are sufficiently large. | ☐ | ☐ | ☐ | ☐ | ☐ |
| Furniture Design | | | | | |
| The classrooms furniture is comfortable. | ☐ | ☐ | ☐ | ☐ | ☐ |
| The classrooms furniture is flexible. | ☐ | ☐ | ☐ | ☐ | ☐ |
| Technical Equipment | | | | | |
| Multiple monitors are installed in the classrooms to allow for the presentation of course content from various angles. | ☐ | ☐ | ☐ | ☐ | ☐ |
| Classrooms can use their computers, etc., to facilitate electronic learning. | ☐ | ☐ | ☐ | ☐ | ☐ |
| Multiple chalkboards (or whiteboards) make learning and communication more effortless in the classroom. | ☐ | ☐ | ☐ | ☐ | ☐ |
| Interactive software is used in the classroom to help with teaching and learning. | ☐ | ☐ | ☐ | ☐ | ☐ |
| There are plenty of power outlets in the classroom. | ☐ | ☐ | ☐ | ☐ | ☐ |
| The classroom has a good WIFI signal. | ☐ | ☐ | ☐ | ☐ | ☐ |

5. Do you have any other requirements or suggestions for the space design of ALCs in HUST?

*Appendix A.2. The Scale of Students' Learning Engagement in ALCs*

Hello! Thank you very much for taking the time to fill out this questionnaire; this research aims to understand your Learning Engagement in ALCs. Your feedback will help to update and improve ALCs. Thank you!

1. Your gender is:

　　　A. Male　　　　　　　　　　　　　　　B. Female

2. Your major is:

　A. Philosophy, Economics, and Law
　B. Education, Literature, and History
　C. Science, Engineering, Agriculture, and Medicine
　D. Military Science, Management, and Art

3. Your classroom number for the course "General English" this semester:
4. Please rate your level of agreement with the following statements as you study in ALCs:

| [1 = Strongly Disagree; 2 = Disagree; 3 = Neutral Attitude; 4 = Agree; 5 = Strongly Agree] | | | | | |
|---|---|---|---|---|---|
| Title | 1 | 2 | 3 | 4 | 5 |
| **Behavioral Dimension** | | | | | |
| I intend to work hard in this class. | ☐ | ☐ | ☐ | ☐ | ☐ |
| When I am in this class, I will pay close attention. | ☐ | ☐ | ☐ | ☐ | ☐ |
| I actively participate in this class's tasks. | ☐ | ☐ | ☐ | ☐ | ☐ |
| In this class, I discuss my ideas with my classmates. | ☐ | ☐ | ☐ | ☐ | ☐ |
| I am excited to participate in group discussions with my classmates in this class. | ☐ | ☐ | ☐ | ☐ | ☐ |
| In this class, I intend to question the teacher actively. | ☐ | ☐ | ☐ | ☐ | ☐ |
| In this class, I will actively discuss my ideas with the teacher. | ☐ | ☐ | ☐ | ☐ | ☐ |
| **Emotional Dimension** | | | | | |
| I get much satisfaction out of studying in this class. | ☐ | ☐ | ☐ | ☐ | ☐ |
| I am interested in what I am learning. | ☐ | ☐ | ☐ | ☐ | ☐ |
| I study diligently because I am eager to learn. | ☐ | ☐ | ☐ | ☐ | ☐ |
| In this class, everyone gets along well. | ☐ | ☐ | ☐ | ☐ | ☐ |
| Students assist one another in class activities and get to know one another. | ☐ | ☐ | ☐ | ☐ | ☐ |
| I enjoy learning with my classmates in this class. | ☐ | ☐ | ☐ | ☐ | ☐ |
| **Cognitive Dimension** | | | | | |
| I try to understand and form my own opinions about the class's content. | ☐ | ☐ | ☐ | ☐ | ☐ |
| During class, I will try to understand the author's intentions. | ☐ | ☐ | ☐ | ☐ | ☐ |
| Outside of class, I will reflect on what I have learned. | ☐ | ☐ | ☐ | ☐ | ☐ |
| I am going to ask questions myself about what I have learned. | ☐ | ☐ | ☐ | ☐ | ☐ |

5. Do you have any additional thoughts or feedback on the relationship between classroom space and student engagement in ALCs?

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
