# Peer review of "The Evaluation of Active Learning Classrooms: Impact of Spatial Factors on Students’ Learning Experience and Learning Engagement"

_sustainability, doi:10.3390/su14084839_

Round 1

Reviewer 1 Report

Hello Dear Authors, After read your work 'The Evaluation of Active Learning Classrooms: Impact of Spa-tial Factors on Students’ Learning Experience and Learning Engagement'  I found some issues to be improve.

Please review the next items>

-The size of figure 2 , looks out of range, also the letter at the images is dificult to read it .

- Review Tables sizes, looks out of range.

-Verify parragraph 5 , the subnumber "5.1 "must not be . 

Author Response

Point 1: The size of figure 2 , looks out of range, also the letter at the images is dificult to read it .

Response 1: The size of figure 2 has been modified, the size of each image in figure 2 has been kept in the same range as the text, and each image in figure 2 has been remade and typeset to meet the clarity required to read the letter in the images.

Point 2: Review Tables sizes, looks out of range.

Response 2: All tables in the text have been adjusted, the size of each table has been kept in the same range as the text.

Point 3: Verify parragraph 5 , the subnumber "5.1 "must not be . 

Response 3: I'm so sorry that I didn't understand the meaning of this point. Is there any need to modify the title of the subnumber " 5.1 ?“ Could you please tell me the content of the problem in detail? thank you very much.

Thank you for your patient and responsible review and answer. Please feel free to contact me if you have any questions.

Reviewer 2 Report

The study presented how the spatial attributes of active learning class ALCs affect students learning experiences and learning engagement. With the evolution and improvements in teaching models, space design, and technology configuration, ALCs have had a tremendous impact on the learning environment of Education for Sustainable Development ESD in terms of theoretical and practical development. This study might successfully increase students’ learning experience and promote learning engagement through empirical research. Students are engaged in the ESD process by rethinking and redesigning these spatial elements to create more effective use of teaching and learning resources as well as innovative use of learning equipment and furniture. This study has certain influence of research time used questionnaires to assess learning experiences and learner engagement. In future studies can be incorporated data collection and analysis, such as classroom observations or qualitative analysis of teachers and students.

The paper content very wide valuable data on teaching models.

General Remarks

Abstract - Please do not use abbreviation in abstract or provide the full name.

Detailed remarks

  1. Lines 29-69. Some references on ALCs and ESD could be added.
  2. Line 218. “…Huazhong University of Science and Technology…” HUST should be added.
  3. Figure 1,2 should be self-explaining. Please add proper information.
  4. Table 1,2 should be self-explaining. Please add proper information.

Author Response

Point 1: Abstract - Please do not use abbreviation in abstract or provide the full name.

Response 1: I used the abbreviation or provide the full name in the abstract because the abstract of this journal is limited to 200 words, as well as the Active Learning Classroom has gradually been known, this classroom has formed a kind of new classroom pattern difference with the traditional classroom, so I marked the abbreviation of "Active Learning Classrooms" as "ALCs" in the first sentence of the abstract, and replaced the subsequent “Active Learning Classrooms” with "ALCs" for to meet the word limit.

Point 2: Lines 29-69. Some references on ALCs and ESD could be added.

Response 2: References on ALCs and ESD have been added between lines 29 and 69 (between lines 29 and 84 in the latest edition).

Point 3: Line 218. “…Huazhong University of Science and Technology…” HUST should be added.

Response 3: " HUST " has been added to line 218. “…Huazhong University of Science and Technology…” ( line 275 in the latest version).

Point 4: Figure 1,2 should be self-explaining. Please add proper information.

Response 4: Figure 1 and Figure 2 have been redrawn. Figure 1 is the conceptual framework diagram of this study, the explanation of this figure has been written in the body of the corresponding paragraph (between lines 261 and 269), so I didn't add additional explanatory information after the picture; All images in Figure 2 have been reformatted and redrawn, and the type of each ALCs have been indicated at the end of the figure.

Point 5: Table 1,2 should be self-explaining. Please add proper information.

Response 5: The analysis software and analysis method of table 1 and Table 2 have been added at the end of the table.

Thank you for your patient and responsible review and answer. Please feel free to contact me if you have any questions.
